# Noninvasive Liver Fibrosis Staging: Comparison of MR Elastography with Extracellular Volume Fraction Analysis Using Contrast-Enhanced CT

**DOI:** 10.3390/jcm11195653

**Published:** 2022-09-25

**Authors:** Keigo Yano, Hiromitsu Onishi, Takahiro Tsuboyama, Atsushi Nakamoto, Takashi Ota, Hideyuki Fukui, Mitsuaki Tatsumi, Takumi Tanigaki, Kunihito Gotoh, Shogo Kobayashi, Keiichiro Honma, Hidetoshi Eguchi, Noriyuki Tomiyama

**Affiliations:** 1Department of Radiology, Osaka University Graduate School of Medicine, Suita 565-0871, Japan; 2Department of Gastroenterological Surgery, Osaka University Graduate School of Medicine, Suita 565-0871, Japan; 3Department of Pathology, Osaka International Cancer Institute, Osaka 541-8567, Japan

**Keywords:** MR elastography, CT, extracellular volume fraction, liver fibrosis

## Abstract

Purpose: To compare the accuracy of liver fibrosis staging with MR elastography and of staging with extracellular volume fraction (fECV) analysis using contrast-enhanced CT. Methods: This retrospective study included 60 patients who underwent both MR elastography and contrast-enhanced CT before liver surgery between October 2013 and July 2020. Two radiologists independently measured liver stiffness of MR elastography and fECV of CT images. Accuracy for liver fibrosis staging was assessed using receiver operating characteristic (ROC) analysis. Correlations between liver stiffness or fECV and liver fibrosis were also evaluated by means of the Spearman rank correlation coefficient. Results: The areas under the ROC curves for MR elastography for each stage differentiation of ≥F1 (0.85, 0.82 for the two radiologists), ≥F2 (0.88, 0.89), ≥F3 (0.87, 0.86), and F4 (0.84, 0.83) were greater than those for fECV analysis with CT (0.64, *p* = 0.06, 0.69, *p* = 0.2; 0.62, *p* < 0.005, 0.63, *p* < 0.005; 0.62, *p* < 0.005, 0.62, *p* < 0.01; and 0.70, *p* = 0.08, 0.71, *p* = 0.2, respectively). The correlation coefficients between liver stiffness and liver fibrosis in A0 (0.67, 0.69 for the two radiologists), A1 (0.64, 0.66) and A2 group (0.58, 0.51) were significantly higher than those between fECV and liver fibrosis (0.28, 0.30; 0.27, 0.31; and 0.23, 0.07; *p* < 0.05 for all comparisons). Conclusion: MR elastography allows for more accurate liver fibrosis staging compared with fECV analysis with CT. In addition, MR elastography may be less affected than fECV analysis by the inflammatory condition.

## 1. Introduction

Liver fibrosis is a wound healing response to chronic liver injury with excessive extracellular matrix deposition caused by various chronic liver disorders such as viral hepatitis and alcoholic liver diseases [1]. The evaluation of liver fibrosis is thus important for assessing the clinical condition of and determining treatment strategies for patients with chronic liver diseases [2,3]. Liver biopsy is currently the gold standard for assessing liver fibrosis, but it has some drawbacks such as invasiveness, observer dependence, and possible sampling errors [4,5,6]. Demand has therefore been growing for noninvasive methods for an accurate evaluation of liver fibrosis. 

Previous studies have reported the usefulness of imaging examinations such as computed tomography (CT) and magnetic resonance (MR) imaging for noninvasively evaluating liver fibrosis [1,2,3,4,7,8,9,10,11,12,13,14,15]. Hepatic extracellular volume fraction (fECV) analysis with contrast-enhanced CT is another noninvasive method for evaluating liver fibrosis. As liver fibrosis progresses, the hepatic extracellular space expands compared to that of normal liver parenchyma [1]. The hepatic fECV can be estimated based on the contrast-enhanced CT images obtained during the equilibrium phase [5,6]. It was found that fECV values correlated significantly with the pathological grades of the liver fibrosis stage [2].

MR elastography, which measures the elastic properties of the liver tissue using dedicated equipment, is yet another noninvasive method to evaluate liver fibrosis. This method shows a strong correlation of elasticity of liver tissue with a degree of fibrosis [16,17,18] and it has been reported that the detection of advanced hepatic fibrosis (stages 3–4) with MR elastography showed a sensitivity of 78% and specificity of 96% [4]. 

Although both fECV and MR elastography have been reported to be useful for assessing liver fibrosis, no previous study has directly compared fECV and MR elastography in terms of diagnostic accuracy for liver fibrosis staging. Comparing which method can more accurately evaluate liver fibrosis in the same subject group seems to have a clinical impact. While fECV analysis can be performed with standard CT scanners, it is necessary to prepare a dedicated device separately from the MR scanner in order to perform MR elastography. Data concerning the accuracy of the two methods may also be required when considering the introduction of the device for MR elastography. Based on histopathological findings, this study thus aimed to compare the diagnostic performance of noninvasive liver fibrosis staging using MR elastography with that using fECV analysis of contrast-enhanced CT.

## 2. Materials and Methods

### 2.1. Study Population

The ethics committee of our institution has approved this study. Eligible for this retrospective study were 110 consecutive patients who underwent liver surgery between October 2013 and July 2020. These patients were referred for clinically indicated contrast-enhanced dynamic CT and contrast-enhanced dynamic MR imaging including MR elastography as common clinical practice before the surgery unless contraindicated. Fifty (46%) of the 110 eligible patients were excluded from the study population because hepatectomy had been performed previously (*n* = 20) because dynamic CT and/or MR elastography was not performed within 100 days prior to surgery (*n* = 26), or because there were no measurable areas on CT and/or MR images (*n* = 4). The remaining 60 patients (41 men and 19 women; mean age, 72 years; range, 67–77 years) constituted the study population (Figure 1). Characteristics and background etiology of the patients are shown in Table 1.

### 2.2. MR Elastography

MR elastography was performed using 3.0-T MR systems (Discovery MR750, Discovery MR750w, or SIGNA Architect; GE Healthcare, Milwaukee, WI, USA). Images provided by MR elastography were acquired using a two-dimensional spin-echo echo-planar sequence. The acquisition parameters comprised repetition time/echo time, 600 msec/62.4–63.4 msec; acquisition matrix size, 64 × 64; section thickness, 10 mm; field of view, 42 cm. The passive drivers transmitted a 60-Hz mechanical vibration. Four slices were obtained at the level of the hilum during a 14-s breath hold. Wave images and stiffness maps were automatically generated.

### 2.3. CT Examinations

Contrast-enhanced dynamic CT was performed using 64-channel (Discovery CT 750 HD; GE Healthcare), 256-channel (Revolution CT; GE Healthcare), or 320-channel (Aquilion ONE; Canon Medical Systems, Otawara, Japan) CT scanners. After scanning the precontrast images, a contrast agent (600 mgI/kg) was intravenously administered with a power injector at a rate of 3–5 mL/s. Images during arterial, portal venous, and equilibrium phases were obtained at approximately 20, 50, and 170 s, respectively, after the CT value had been enhanced to 100 Hounsfield units (H.U.) in the aorta at the hepatic hilum level.

### 2.4. Image Assessments

Two radiologists (with five years (K.Y.) and 21 years of experience respectively (H.O.)) independently measured liver stiffness shown on MR elastographs and attenuation on CT images for fECV analyses. Both radiologists were blinded to the histopathology findings and intra-operative findings.

For MR elastography, liver stiffness was measured with the standard procedure by placing free-hand ROIs on the right lobe, avoiding lesions, large blood vessels, and inappropriate regions such as hot/cold spots on the stiffness map [19] (Figure 2). The values for four slices were averaged for each patient. The MR elastography algorithm had been changed at our institution from multiscale direct inversion (MSDI) to multimodel direct inversion (MMDI) midway through the research period. Liver stiffness measured by MMDI is reportedly approximately 7% lower than that measured by MSDI [20]. Accordingly, the values obtained with MSDI (*n* = 11) were corrected to correspond to the values for MMDI.

Approximately 1-cm^2^ regions of interest (ROIs) were placed in the lateral, anterior, and posterior segments, avoiding blood vessels, bile ducts, and focal lesions, on pre-contrast CT images and equilibrium phase CT images (Figure 2) to obtain the degree of enhancement of liver parenchyma (*E_liver_*). For the measurement of the enhancement values of the aorta (*E_aorta_*), the ROIs were drawn as large as possible on the lumen of the aorta without covering the vessel wall. To assess whether the scan timing of CT images obtained during the equilibrium phase was appropriate, the enhancement values of the main trunk of the portal vein were also measured and compared with those of the aorta [1]. Hepatocyte fECV was calculated using the following equation:fECV = *E_liver_* × (100 – [Haematocrit])/*E_aorta_*
(1)
where [Haematocrit] represents the haematocrit values (%) obtained with the blood test performed immediately before the CT examinations.

### 2.5. Histologic Analysis

Liver fibrosis of specimens obtained by hepatectomy was evaluated by a pathologist using a 5-step scale according to the METAVIR scoring system: F0, no fibrosis; F1, portal septal fibrosis; F2, portal fibrosis and a few septa; F3, multiple septa without cirrhosis; F4, cirrhosis [21]. Liver inflammation was also graded using a 4-step scale: A0, no activity; A1, mild activity; A2, moderate activity; A3, severe activity. The pathologist was blinded to MR and CT findings.

### 2.6. Statistical Analysis

Correlations between liver stiffness on MR elastographs or fECV on CT images and pathological liver fibrosis stage were evaluated by using the Spearman rank correlation coefficient. Subgroups classified according to pathological inflammation grade were also assessed. These correlation coefficients were also statistically compared using the Fisher z transformation method [22].

The accuracy of MR elastography and of fECV analysis using CT for fibrosis staging was assessed with receiver operating characteristic (ROC) analysis. The areas under the ROC curve (AUROC) were calculated and compared using the method of DeLong et al. for the following stages: stage F0 versus F1–4, F0–1 versus F2–4, F0–2 versus F3–4, and F0–3 versus F4. Sensitivity and specificity for each stage determined with either technique was also determined. We also compared AUROCs for fibrosis staging by fECV measurements using CT and MR elastography for the patients who underwent resection in the right lobe (*n* = 38). At this time, the analysis was performed using only the ROIs placed on the right lobe on the CT images. Moreover, we performed subgroup analyses for the patient group who underwent non-anatomical resection (e.g., partial resection, *n* = 25) and the patient group who underwent anatomical resection (e.g., lobectomy, segmentectomy, or sub-segmentectomy, *n* = 35) in order to investigate the presence or absence of effects on the liver parenchyma due to tumor compression.

The interobserver reliabilities of the two radiologists were evaluated with the interclass correlation coefficient (ICC). The ICC criteria were as follows: poor relationship (ICC < 0.40), fair-to-good relationship (ICC = 0.40–0.75), and excellent relationship (ICC > 0.75). All statistical analyses other than the Fisher z transformation method were performed with EZR (Saitama Medical Center, Jichi Medical University, Saitama, Japan), which is a graphical user interface for R (The R Foundation for Statistical Computing, Vienna, Austria, version 4.0.3) [23]. P values less than 0.05 were considered significant.

## 3. Results

The number of patients for each pathological liver fibrosis stage was: F0, *n* = 12; F1, *n* = 13; F2, *n* = 10; F3, *n* = 8; and F4, *n* = 17. The interval between CT examinations and surgery ranged from 2 to 100 days (median 32.5 days) and the interval between MR examinations and surgery ranged from 1 to 97 days (median 29 days). The differences in CT values between the aorta and the portal vein trunk were 10 H.U. or less in all cases. Correlation of liver stiffness obtained using MR elastography with liver fibrosis stage (*rs* = 0.70 and 0.68) was significantly superior to that of fECV analysis using CT (*rs* = 0.28 and 0.31, *p* < 0.01, for either radiologist) (Figure 3). The AUROCs for MR elastography of staging F0–1 versus F2–4 and F0–2 versus F3–4 were significantly greater than those for fECV analysis using CT for both radiologists (*p* < 0.01) (Figure 4, Table 2). The AUROCs for MR elastography of staging F0 versus F1-4 and F0-3 versus F4 were also greater than those for fECV analysis using CT, although the difference did not reach statistical significance. There were no significant differences in sensitivity for any staging between the two techniques (Table 3). The specificity of MR elastography was significantly greater than that of fECV analysis using CT for F0–1 versus F2–4, F0–2 versus F3–4, and F0–3 versus F4 for one radiologist (Table 3).

The rate of concordance between the two radiologists showed high values overall and excellent ICC values in particular for liver stiffness (ICC = 0.96) and fECV (ICC = 0.89).

The number of patients for each liver inflammation grade was: A0, *n* = 17; A1, *n* = 28; A2, *n* = 15; and A3; *n* = 0. The correlation coefficients between liver stiffness and liver fibrosis for MR elastography in A0, A1, and A2 groups were significantly higher than those between liver stiffness and liver fibrosis for fECV analysis using CT and liver fibrosis for any of the comparisons (Table 4).

The AUROC values for the patients who underwent resection in the right lobe are shown in Table 5. The results of subgroup analyses are shown in Table 6.

## 4. Discussion

Our findings indicate that both MR elastography and fECV analysis with CT were helpful for determining the severity of liver fibrosis. Of the two methods, diagnostic accuracy (i.e., AUROC) for fibrosis staging with MR elastography was greater than that for fECV analyses for every stage. Our results for MR elastography showed approximately the same diagnostic accuracy as results reported in previous articles [4,19]. Regarding fECV analysis with CT, two previous studies have been reported [14,15]. In one study, the AUROC results were superior to our results [14], and in the other, they were basically similar to our results [15]. Our results also showed that the correlation of liver stiffness obtained by means of MR elastography with the liver fibrosis stage was significantly superior to that for fECV analysis using CT. Thus, MR elastography may be preferable for the evaluation of liver fibrosis when both methods are clinically available. However, MR elastography is only marginally better than other non-invasive tests (e.g., serum markers, transient elastography, and shear wave elastography) for F3–F4 fibrosis, although it is the most accurate non-invasive method for staging liver fibrosis [24]. Therefore, according to the EASL Clinical Practice Guidelines, MR elastography is not recommended as a first-line non-invasive test given its cost and limited availability [24]. In addition, MR elastography has the following drawbacks. Although rare, it may not be possible to measure liver fibrosis with MR elastography because of insufficient penetration of the vibration wave into the liver [25]. In addition, MR elastography is not suitable for patients after the right lobectomy of the liver since measurement of liver stiffness in the left lobe has not been established [26]. Hence, fECV analysis with CT may be preferable to MR elastography in certain instances.

The subgroup analyses showed almost no correlation between fECV and the fibrosis stage for inflammation grade A2 (i.e., correlation coefficients of 0.25 or less), while there were moderate correlations between liver stiffness determined with MR elastography and fibrosis stage for all the other inflammation grades. To our knowledge, no study has yet been conducted to determine whether liver inflammation affects the diagnostic accuracy of liver fibrosis by fECV with CT. Although fibrosis is the major cause of extracellular expansion, edema and inflammation may also cause extracellular expansion [27]. If inflammation and fibrosis coexist, fECV shows a combination of the two, but it cannot discriminate between them [28]. Thus, inflammation could interfere with the assessment of fibrosis by fECV, and fECV may be a good indicator of the fibrosis stage only in the absence of significant inflammation.

In the pathological analysis, the fibrosis was evaluated at the liver parenchyma relatively close to the liver tumor since the resected specimens were used. Hence the liver parenchyma compressed by the tumors may have been assessed, especially when the patient had undergone non-anatomical resection (e.g., partial resection). In the meanwhile, clear areas of the liver away from the tumor were assessed for MR elastography and fECV with CT. It may be possible that the effect of tumor compression on the pathological evaluation affects the results in the present study. The results of subgroup analyses for the patients who underwent non-anatomical resection (e.g., partial resection) and the patient who underwent anatomical resection (e.g., lobectomy, segmentectomy, or sub-segmentectomy) were similar to the results for all patients. This suggests that the compression by the tumors did not significantly affect the pathological evaluation in this study.

In previously reported articles on fECV analyses, the scan timing of the equilibrium phase on contrast-enhanced dynamic CT varied. In some studies, the equilibrium phase images were obtained 10 min or more after the start of contrast media administration [5,6]. However, in other studies, they were obtained after 180 s and the liver fibrous stage could be accurately estimated [1,2]. In our study, scanning was performed 170 s after the bolus-tracking threshold, which is comparable to approximately 190 s after the beginning of contrast media administration. The differences in CT values in our findings between the aorta and the portal vein trunk were 10 H.U. or less in all cases, indicating that the equilibrium phase was obtained properly [1]. A shorter scan delay may be considered more manageable for daily clinical practice.

Our research has certain limitations. First, our study was a single-center and retrospective study. Second, the time interval between surgery, CT examination, and MR elastography examination were as long as 100 days. If the disease progresses between image examination and surgery, the degree of fibrosis may be affected. The third limitation is the anatomical inconsistency between the sites where the fECV values or liver stiffness were measured on CT or MR images and where fibrosis stages were evaluated pathologically. As for MR elastography, it is recommended that the ROIs are placed on the right lobe of the liver to avoid the effect on the heartbeat [25]. In our study, both fECV and MRE were measured in accordance with previously reported methods [2,19]. In order to match the site of evaluation, we also added the analysis in which the subjects were limited to patients who underwent resection in the right lobe. In the analysis, MR elastography, fECV with CT, and pathology were all evaluated in the right lobe and a similar finding was observed in the results.

In conclusion, MR elastography allows for more accurate liver fibrosis staging compared with fECV analysis using CT. In addition, MR elastography may be less affected than fECV analysis by the inflammatory condition.

## Figures and Tables

**Figure 1 jcm-11-05653-f001:**
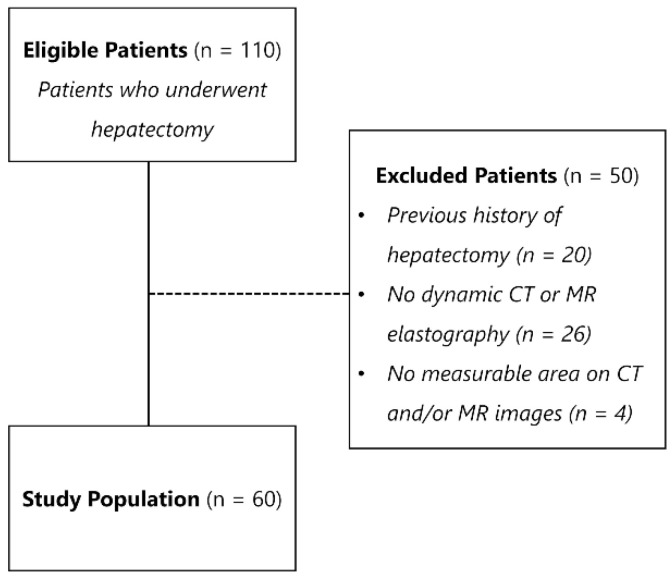
Flowchart of patient enrollment.

**Figure 2 jcm-11-05653-f002:**
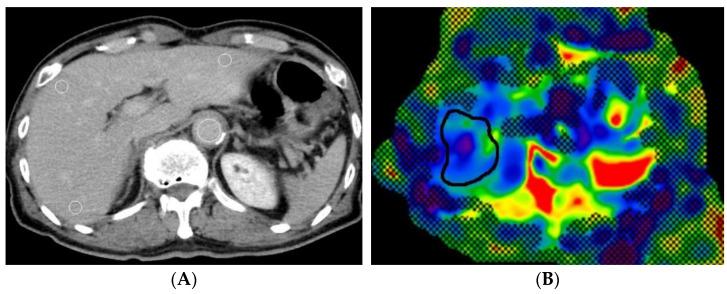
A 77-year-old man with liver fibrosis stage F1 (**A**,**B**) and an 82-year-old man with liver fibrosis stage F3 (**C**,**D**). Circular ROIs were placed in the liver and the aorta on contrast-enhanced CT during the equilibrium phase to analyze extracellular volume fraction (**A**,**C**). Free-hand ROIs were drawn in the right lobe of the liver on an MR elastogram to measure liver stiffness (**B**,**D**).

**Figure 3 jcm-11-05653-f003:**
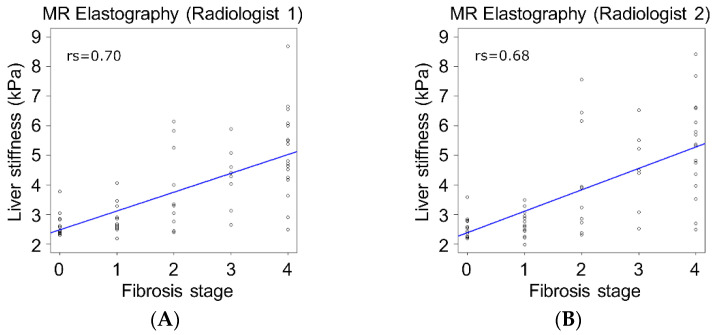
These graphs show the correlation between liver fibrosis stage and liver stiffness obtained with MR elastography (**A**,**B**) or hepatic extracellular volume fraction (fECV) analysis obtained with CT (**C**,**D**). The correlation of liver stiffness at MR elastography with liver fibrosis stage was significantly superior to that of fECV analysis at CT for both radiologists (*p* < 0.01).

**Figure 4 jcm-11-05653-f004:**
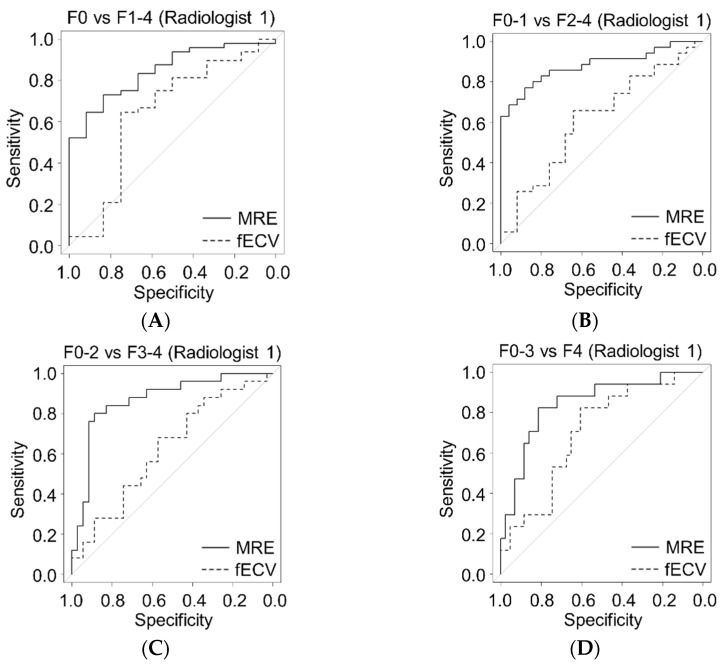
These graphs are ROC curves for the diagnostic accuracy of liver fibrosis staging with liver stiffness of MR elastography and fECV of dynamic CT for two radiologists (**A**–**H**). The AUROCs for MR elastography were significantly higher than those for fECV analysis with CT for both radiologists in the staging F0–1 versus F2–4 (**B**,**F**) and F0–2 versus F3–4 (C, G) (*p* < 0.01).

**Table 1 jcm-11-05653-t001:** Patients’ demographics.

Characteristic	Numbers
Age (years) *	72 (67–77)
Gender	
Male	41/60 (68.3%)
Female	19/60 (31.7%)
Liver fibrosis stage	
F0	12/60 (20.0%)
F1	13/60 (21.7%)
F2	10/60 (16.7%)
F3	8/60 (13.3%)
F4	17/60 (28.3%)
Liver inflammation grade	
A0	17/60 (28.3%)
A1	28/60 (46.7%)
A2	15/60 (25.0%)
A3	0/60 (0.0%)
Liver disease and etiology	
None	13/60 (21.7%)
Alcoholic liver disease	7/60 (11.7%)
Nonalcoholic steatohepatitis	5/60 (8.3%)
Autoimmune hepatitis	1/60 (1.7%)
Viral hepatitis type B	6/60 (10.0%)
Viral hepatitis type C	21/60 (35.0%)
Viral hepatitis type B & C	2/60 (3.3%)
Cryptogenic chronic hepatitis	5/60(8.3%)
Pathology of liver masses	
Hepatocellular carcinoma	48/60 (80.0%)
Intrahepatic cholangiocellular carcinoma	4/60 (6.7%)
Combined hepatocellular carcinoma	4/60 (6.7%)
Others	4/60 (6.7%)
Type of Hepatectomy	
Lobectomy	12/60 (20.0%)
Segmentectomy	18/60 (30.0%)
Sub-segmentectomy	5/60 (8.3%)
Partial hepatectomy	25/60 (41.7%)

Note: Unless otherwise specified, data show numbers of patients, with percentages in parentheses. * Data show the median, with a range in parentheses.

**Table 2 jcm-11-05653-t002:** Diagnostic accuracy of liver fibrosis staging with MR elastography and fECV analysis using CT.

Liver Fibrosis Stage	AUROC		*p* Value
	MR Elastography	fECV Using CT	
Radiologist 1			
F0 versus F1–4	0.85	0.64	0.06
F0–1 versus F2–4	0.87	0.62	<0.005
F0–2 versus F3–4	0.87	0.62	<0.005
F0–3 versus F4	0.84	0.70	0.08
Radiologist 2			
F0 versus F1–4	0.82	0.69	0.2
F0–1 versus F2–4	0.89	0.63	<0.005
F0–2 versus F3–4	0.86	0.62	<0.01
F0–3 versus F4	0.83	0.71	0.2

Note: AUROC = area under the receiver operating characteristic curve, fECV = extracellular volume fraction.

**Table 3 jcm-11-05653-t003:** Sensitivities and specificities of liver fibrosis staging obtained with MR elastography and fECV analysis using CT.

Liver Fibrosis Stage	Sensitivity			Specificity		
	MRE	fECV	*p* Value	MRE	fECV	*p* Value
Radiologist 1						
F0 versus F1–4	0.67	0.65	1.00	0.83	0.75	1.00
F0–1 versus F2–4	0.74	0.66	0.61	0.88	0.64	0.11
F0–2 versus F3–4	0.80	0.68	0.51	0.89	0.57	<0.01
F0–3 versus F4	0.82	0.82	1.00	0.81	0.58	<0.05
Radiologist 2						
F0 versus F1–4	0.71	0.65	0.66	0.92	0.75	0.62
F0–1 versus F2–4	0.71	0.54	0.24	1.00	0.76	<0.05
F0–2 versus F3–4	0.80	0.60	0.23	0.91	0.71	0.07
F0–3 versus F4	0.82	0.71	0.68	0.79	0.70	0.45

Note: MRE = magnetic resonance elastography, fECV = extracellular volume fraction.

**Table 4 jcm-11-05653-t004:** Correlations between liver stiffness determined with MR elastography or with fECV using CT and pathological liver fibrosis stage including subgroup analyses.

Spearman Rank Correlation Coefficient			
	MR Elastography	fECV of CT	*p* Value
Radiologist 1			
All (*n* = 60)	0.70	0.28	<0.01
Inflammation grade A0 (*n* = 17)	0.67	0.28	<0.01
Inflammation grade A1 (*n* = 28)	0.64	0.27	<0.05
Inflammation grade A2 (*n* = 15)	0.58	0.23	<0.05
Radiologist 2			
All (*n* = 60)	0.68	0.31	<0.01
Inflammation grade A0 (*n* = 17)	0.69	0.30	<0.005
Inflammation grade A1 (*n* = 28)	0.66	0.31	<0.05
Inflammation grade A2 (*n* = 15)	0.51	0.07	<0.01

Note: fECV = extracellular volume fraction.

**Table 5 jcm-11-05653-t005:** Diagnostic accuracy of liver fibrosis staging with MR elastography and fECV analysis using CT for the patients who underwent resection in the right lobe.

Liver Fibrosis Stage	AUROC		*p* Value
	MR Elastography	fECV Using CT	
Radiologist 1			
F0 versus F1–4	0.96	0.68	0.11
F0–1 versus F2–4	0.85	0.46	<0.005
F0–2 versus F3–4	0.8	0.55	<0.01
F0–3 versus F4	0.78	0.57	<0.01
Radiologist 2			
F0 versus F1–4	0.92	0.59	<0.01
F0–1 versus F2–4	0.87	0.51	<0.005
F0–2 versus F3–4	0.79	0.59	0.13
F0–3 versus F4	0.78	0.51	0.06

Note: AUROC = area under the receiver operating characteristic curve, fECV = extracellular volume fraction.

**Table 6 jcm-11-05653-t006:** Diagnostic accuracy of liver fibrosis staging with MR elastography and fECV analysis using CT in subgroup analyses.

Liver Fibrosis Stage	AUROC		*p* Value
	MR Elastography	fECV Using CT	
Non-anatomical resection group			
Radiologist 1			
F0 versus F1–4	N/A	N/A	N/A
F0–1 versus F2–4	0.89	0.67	0.17
F0–2 versus F3–4	0.83	0.78	0.73
F0–3 versus F4	0.84	0.81	0.8
Radiologist 2			
F0 versus F1–4	N/A	N/A	N/A
F0–1 versus F2–4	0.91	0.77	0.27
F0–2 versus F3–4	0.83	0.75	0.57
F0–3 versus F4	0.84	0.79	0.68
Anatomical resection group			
Radiologist 1			
F0 versus F1–4	0.82	0.64	0.16
F0–1 versus F2–4	0.88	0.61	<0.05
F0–2 versus F3–4	0.91	0.52	<0.005
F0–3 versus F4	0.89	0.71	0.3
Radiologist 2			
F0 versus F1–4	0.8	0.68	0.46
F0–1 versus F2–4	0.88	0.56	<0.05
F0–2 versus F3–4	0.91	0.5	<0.005
F0–3 versus F4	0.84	0.65	0.39

Note: AUROC = area under the receiver operating characteristic curve, fECV = extracellular volume fraction, N/A = not available (analysis results were not calculated due to small data sets).

## Data Availability

Not applicable.

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
