# Peer review of "Noninvasive Liver Fibrosis Staging: Comparison of MR Elastography with Extracellular Volume Fraction Analysis Using Contrast-Enhanced CT"

_jcm, 2022, doi:10.3390/jcm11195653_

Round 1
Reviewer 1 Report
This is an interesting study comparing the accuracy of MR elastography and fECV to detect and stage liver fibrosis. The authors report that MRE is better compared to fECV.
1. Were the reporting radiologists blinded to intra-operative findings?
2. Was the pathologist blinded to MR and CT
3. There may be a methodological problem here as fibrosis would have been checked in the resected liver which may be in close proximity and hence compressed by the tumor- especially when the patient had undergone non-anatomical resection while the MRE and fECV would have been measured in clear areas of the liver away from the tumor. The authors should provide information regarding the type of hepatectomy- anatomical and non-anatomical performed in these patients. It may also be useful to analyse these two groups separately to see if the current findings still hold true.
Author Response
Manuscript ID: jcm-1906199
Title: Noninvasive Liver Fibrosis Staging: Comparison of MR Elastography with Extracellular Volume Fraction Analysis Using Contrast-Enhanced CT
Response to Reviewer 1
Dear reviewer:
We would like to thank reviewers for their very helpful and valuable suggestions. We have tried our best to revise the manuscript and feel the manuscript is very much improved.
All the comments were accepted and carefully revised. Below, we provide detailed responses to each of the points raised by the reviewers.
- Were the reporting radiologists blinded to intra-operative findings?
Yes, they were blinded to intra-operative findings. We have added a note about this in the text.
- Was the pathologist blinded to MR and CT
Yes, they were blinded to MR and CT images and reports. We have added a sentence about this in the text.
- There may be a methodological problem here as fibrosis would have been checked in the resected liver which may be in close proximity and hence compressed by the tumor- especially when the patient had undergone non-anatomical resection while the MRE and fECV would have been measured in clear areas of the liver away from the tumor. The authors should provide information regarding the type of hepatectomy- anatomical and non-anatomical performed in these patients. It may also be useful to analyse these two groups separately to see if the current findings still hold true.
According to the reviewer’s suggestion, we have added the information regarding the type of hepatectomy in Table 1. At that time, we noticed that some of the percentage figures in the table were inaccurate, so we corrected them. We apologize for not being able to confirm.
We have also added the subgroup analyses for the non-anatomical and anatomical patient groups. In the subgroup analyses, we showed the results similar to the original analysis.
Thank you.
Reviewer 2 Report
I am highly interesting by this paper. However, I am a ittle disappointed by the description of the liver specimens of the histologic analysis. The liver is often heterogeneous in terms of fibrosis or inflammation and it would be interesting to compare the same regions of liver by MRI, Scann and histologic analysis. For F1,2,3, MRI is peerhaps more interesting than histologic analysis because it gets a general evaluation of the liver and not a unique specimen.
Author Response
Manuscript ID: jcm-1906199
Title: Noninvasive Liver Fibrosis Staging: Comparison of MR Elastography with Extracellular Volume Fraction Analysis Using Contrast-Enhanced CT
Response to Reviewer 2
Dear reviewer:
We would like to thank reviewers for their very helpful and valuable suggestions. We have tried our best to revise the manuscript.
All the comments were accepted and carefully revised. Below, we provide detailed responses to each of the points raised by the reviewers.
“I am highly interesting by this paper. However, I am a little disappointed by the description of the liver specimens of the histologic analysis. The liver is often heterogeneous in terms of fibrosis or inflammation and it would be interesting to compare the same regions of liver by MRI, Scann and histologic analysis.”
We fully understand that the reviewer's concern is the most important point in the present study. As a response to this, we have added the analysis in which the subjects were limited to patients who underwent resection in the right lobe, in order to match the site of evaluation. In the analysis, we showed the results similar to the original analysis.
Thank you.